# Bangla MedER: Multi-BERT ensemble approach for the recognition of Bangla medical entity

**Tanjim Taharat Aurpa**[1]*, **Farzana Akter**[2]*, **Md. Mehedi Hasan**[2], **Shakil Ahmed**[2], **Shifat Ara Rafiq**[3], **Fatema Khan**[4], **Rubel Sheikh**[5]

**1** Department of Data Science and Engineering, University of Frontier Technology, Bangladesh, Gazipur, Bangladesh, **2** Department of IoT and Robotics Engineering, University of Frontier Technology, Bangladesh, Gazipur, Bangladesh, **3** Department of Software Engineering, University of Frontier Technology, Bangladesh, Gazipur, Bangladesh, **4** Department of Computer Science and Engineering, University of Liberal Arts Bangladesh, Dhaka, Bangladesh, **5** Department of Educational Technology and Engineering, University of Frontier Technology, Bangladesh, Gazipur, Bangladesh

* aurpa0001@uftb.ac.bd (TTA); farzana0001@uftb.ac.bd (FA)

## Abstract

Medical Entity Recognition (MedER) is an essential NLP task for extracting meaningful entities from the medical corpus. Nowadays, MedER-based research outcomes can remarkably contribute to the development of automated systems in the medical sector, ultimately enhancing patient care and outcomes. While extensive research has been conducted on MedER in English, low-resource languages like Bangla remain underexplored. Our work aims to bridge this gap. For Bangla medical entity recognition, this study first examined a number of transformer models, including BERT, DistilBERT, ELECTRA, and RoBERTa. We also propose a novel Multi-BERT Ensemble approach that outperformed all baseline models with the highest accuracy of 89.58%. Notably, it provides an 11.80% accuracy improvement over the single-layer BERT model, demonstrating its effectiveness for this task. A major challenge in MedER for low-resource languages is the lack of annotated datasets. To address this issue, we developed a high-quality dataset tailored for the Bangla MedER task. The dataset was used to evaluate the effectiveness of our model through multiple performance metrics, demonstrating its robustness and applicability. Our findings highlight the potential of Multi-BERT Ensemble models in improving MedER for Bangla and set the foundation for further advancements in low-resource medical NLP.

## 1 Introduction

NLP (Natural Language Processing) techniques have become popular in medical research day by day, enabling efficient management and analysis of textual medical data. One of the key techniques of NLP is Named Entity Recognition (NER), which focuses on identifying specific terms in text, such as diseases, medications, and symptoms. In the healthcare industry, where vast amounts of unstructured medical text including clinical notes, research papers, and electronic health records need to

**Data availability statement:** To conduct this research, we have collected corpus from Bangla public medicine banks and medical blog posts and manually annotated them. No human information, private social media data, or animal research has been included during the data collection process. The annotated Bangla MedER dataset is publicly available on Kaggle Data Source: https://www.kaggle.com/datasets/tanjimtaharataurpa/bangla-medical-entity-dataset.

**Funding:** The author(s) received no specific funding for this work.

**Competing interests:** The authors have declared that no competing interests exist.

be processed, NER plays a crucial role in improving research efficiency and decision making. Medical Entity Recognition (MedER) is a specialized form of NER introduced specifically for the medical domain.

While English-language medical NER has reached high performance levels through large-scale datasets and transformer-based models, the vast majority of the world's languages—particularly low-resource ones like Bangla—remain severely underserved. Existing Bangla NER systems, primarily designed for general-domain text, exhibit substantial performance degradation when applied to medical documents due to complex clinical terminology, extensive code-switching with English medical terms, morphological richness, and the frequent use of abbreviated or colloquial expressions for symptoms and diseases. Given these limitations, there is a pressing need to design a domain-adapted, transformer-based MedER framework specifically for Bangla, supported by a high-quality annotated dataset.

Despite the rapid global advancement of biomedical NLP, Bangla remains critically underrepresented in medical NER research. The existing literature shows that most prior Bangla NER studies focus on general-domain applications and lack domain-specific linguistic coverage, limiting their applicability to real medical contexts. As highlighted in the reviewed works, there is no large-scale, high-quality Bangla medical NER dataset, and current models fail to effectively capture clinical terminology, English–Bangla code-mixing, and diverse morphological patterns that frequently appear in medical narratives. Furthermore, state-of-the-art biomedical models available in English—such as BioBERT, ClinicalBERT, and PubMedBERT—benefit from extensive in-domain pretraining corpora, whereas Bangla has no equivalent medically pretrained transformer model. This shortage of resources, combined with the complexity of Bangla medical expressions, makes accurate medical entity extraction extremely challenging. Therefore, a dedicated Bangla MedER system supported by a domain-specific dataset and enhanced through ensemble transformer architectures is urgently needed to improve clinical text understanding, support medical decision-making, and enable the development of robust healthcare NLP applications for Bangla-speaking populations.

Our aim is to recognize medical entities in Bangla medical texts. This type of task helps to retrieve necessary information, like medicine names, disease names, etc., efficiently from large amounts of medical data in a short amount of time. In addition, finding symptoms and writing prescriptions will be easier for people with medical interests. It contributes to building automated medical systems by accurately identifying and classifying medical entities, such as treatment names, symptoms, and medical conditions, from unstructured medical data. Moreover, by maneuvering MedER, healthcare systems can be efficient in complex medical information processing, improving patient care and decision-making for better research insights.

Accurate MER is crucial in numerous applications, including clinical decision support systems, systematic reviews, and epidemiological studies. It enables the effective retrieval of relevant patient information, improves the efficiency of research processes, and supports the integration of data across different healthcare systems. Traditional approaches include rule-based methods that utilize predefined patterns and dictionaries for entity identification. However, machine learning and deep

learning techniques have demonstrated superior performance by allowing systems to learn from annotated corpora, resulting in more accurate and context-aware recognition. Recent advances in deep learning, particularly models like BERT (Bidirectional Encoder Representations from Transformers), have significantly improved NLP tasks, including entity recognition [1,2]. BERT's ability to learn word context from large text datasets makes it highly effective in understanding the meaning of medical terms, even in complex or informal texts.

Variants such as ClinicalBERT, BioBERT, and PubMedBERT have demonstrated remarkable gains in English biomedical NER by leveraging large-scale pretraining on in-domain corpora. For Bangla, several monolingual and multilingual BERT models (Bangla-BERT, XLM-RoBERTa, mBERT, etc.) have been released, but none were pretrained specifically on Bangla biomedical text, resulting in suboptimal representation of medical concepts. Moreover, single-model approaches often suffer from inherent biases and limitations of individual pretraining corpora, prompting growing interest in ensemble techniques that combine complementary strengths of multiple models to achieve higher robustness and accuracy. Unlike English, Bangla medical texts often contain specialized vocabulary, informal expressions, and inconsistent terminology. In addition, the lack of large annotated datasets makes it difficult to retrieve important information.

This research aims to address these challenges by leveraging BERT for Medical Entity Recognition in Bangla medical text. We introduce Bangla MedER, a Multi-BERT Ensemble framework, and fine-tune the ensemble BERT on a newly developed dataset of Bangla medical statements to identify key entities such as medicine names, disease names, and common medical terms. In doing so, our work contributes to bridging the resource gap in medical NLP in Bangla and opens new possibilities for practical applications, including automated record management, diagnosis support, and improved access to healthcare information for Bangla-speaking communities.

### 1.1 Research objective

This study aims to develop an innovative approach for identifying medical entities in Bangla text, such as medicine names, disease names, and other healthcare-related terms. By leveraging transformer-based models like BERT and creating a novel dataset of real-world Bangla medical statements, this research seeks to overcome language-specific challenges and advance the field of medical entity recognition. The primary objectives of this study are

- To design a method for accurately identifying key entities such as medicine names, disease names, and treatment-related terms in Bangla medical texts.
- To fine-tune BERT and its variations to handle the complexities of Bangla medical language, including diverse vocabulary and domain-specific terminology.
- To develop and annotate a high-quality dataset of real-world Bangla medical statements focused on medicines and diseases.
- To evaluate the model's performance using metrics such as Accuracy, Loss, Macro Average F1 Score, and Micro Average F1 score, benchmarking it against existing state-of-the-art models.

The rest of the paper is organized as follows: Sect 2 reviews existing research on medical entity recognition in low-resource languages like Bangla, Sect 3 outlines the entity recognition methodology, Sect 4 represents the result analysis, Sect 5 addresses the challenges and their implications in real-world applications, Sect 6 discuss the conclusion.

## 2 Related work

Medical Entity Recognition (MedER) has seen significant advancements with transformer-based models. To enhance Bangla NER, Ashrafi et al. [2] introduced BANNER, a BERT embedding-based model with BiLSTM and Conditional Random Field (CRF) layers. By utilizing a cost-sensitive loss function to address class imbalances, BANNER significantly outperformed previous models, achieving a macro F1 score of 65.96%, micro F1 of 90.64%, and MUC F1 of 72.04%, an

8% improvement over earlier models. To address privacy concerns in collecting labeled medical data, Ge et al. [3] proposed FedNER, a federated learning-based framework. FedNER consistently outperformed other models across benchmark datasets, achieving the highest F1 scores while preserving data privacy. A comprehensive review by Perera et al. [4] analyzed various BioNER techniques, including rule-based, machine learning, and deep-learning methods like BERT and BioBERT. They highlighted challenges such as non-standard abbreviations, synonym variations, and polarity analysis in entity recognition. Most MedER research has been conducted in English, with limited resources for other languages. Miranda et al. [5] introduced CANTEMIST (CANcer TExt MIning Shared Task) at IberLEF 2020, a Spanish-language initiative for cancer text mining. Using deep learning models like LSTM, BERT, and BETO, they developed the Cantemist corpus with tumor morphology entities linked to Spanish ICD-O codes. In the Italian language, Catelli et al. [6] introduced the SIRM COVID-19 dataset from the Italian Society of Radiology. They evaluated Bi-LSTM+CRF and BERT models using the English i2b2 2014 de-identification corpus. Their best-performing approach used sequential training with Bi-LSTM+CRF and multilingual embeddings, outperforming other strategies. A neural network-based NER approach in Electronic Health Records (EHRs) was developed by Gligic et al. [7]. Using transfer learning, their model achieved an F1 score of 94.6%, demonstrating the effectiveness of deep learning in clinical text mining. Liu et al. [8] introduced Med-BERT, which improved traditional BERT models by integrating a span flat-lattice transformer, enhancing recognition accuracy and privacy handling. Med-BERT achieved the highest F1 scores of 0.93 and 0.69 on CMR and CMEEE datasets, respectively. Similarly, French et al. [9] provided an overview of Biomedical Entity Linking (BEL) systems, analyzing 60 works from 1980 to 2022, identifying key challenges, datasets, and future research directions. Bangla NER research has been gaining traction, addressing the scarcity of annotated datasets. Alvi et al. [10] introduced the BNER dataset, containing eight entity types. For Bangla Medical NER, Muntakim et al. [11] developed a dataset containing 117,000 tokens, categorized into three key classes: Chemicals and Drugs (CD), Disease and Symptom (DS), and Anatomy (ANAT) in IOB2 format. Their best-performing model, Bi-LSTM-CRF, achieved 98% accuracy and a macro F1 score of 78%. Lima et al. [12] implemented a Bangla NER system on a dataset containing 1 million tokens across six entity types. They developed two models: type-1 (word-level + character-level features) and type-2 (word-level only). The BGRU+CNN+CRF model with Word2Vec (CBOW) embeddings achieved the highest F1 score of 92.31%. In Complex Named Entity Recognition (CNER), Shahgir et al. [13] used the BanglaCoNER dataset to compare CRF models with BanglaBERT. Their fine-tuned BanglaBERT model achieved the highest F1 score of 0.79. Their model, NR/IndicbnBERT, achieved the highest macro F1 score of 86%, while cross-dataset validation showed a micro F1 score of 74%, indicating its impact on Bangla NLP. A specialized dataset for Mathematical Entity Recognition (MER) in Bangla was created by Aurpa et al. [14], containing 13,717 instances classified as numbers, operators, and common mathematical terms. When applying mBERT, they achieved 99.76% accuracy. Expanding on this, the authors later introduced an ensemble BERT architecture, combining two BERT layers with different input sequences, further improving performance [15]. A new Gazetteer containing over 96,000 entities was developed by Farhan et al. [16]. By integrating Gazetteer with CRF and BanglaBERT embeddings, they achieved an F1 score of 0.8267, outperforming traditional deep learning methods. Barragán et al. [17] compare a traditional BERT-based NER model with a few-shot GPT approach using a Spanish EHR dataset. GPT slightly outperforms BERT (F1: 0.95 vs. 0.94), though BERT offers faster, millisecond-level inference suitable for real-time use. GPT requires heavier computational resources but provides stronger generalization, lower data dependency, and effective zero-shot capabilities, making it appealing for rapid deployment across diverse medical domains. Yan et al. [18] employ RoBERTa-wwm-ext-GCA-CRF for Chinese medical NER, achieving F1-scores of 91.90% and 64.36%. While effective for Chinese texts, its cross-lingual adaptability remains untested. The architecture's multi-head attention and BiLSTM components increase computational cost, suggesting the need for lightweight models, cross-lingual pretraining, and knowledge-graph integration to improve scalability and generalization. Wei et al. [19] present a hybrid framework combining medical ontologies with deep learning, enhancing context-aware entity recognition. Leveraging self-supervised learning and domain

adaptation, their approach adapts well to evolving terminology without extensive manual annotation. Sun et al. [20] propose a lexicon-enhanced BERT model using a Lexicon Adapter, Star-Transformer, BiLSTM, and RoPE to improve boundary detection in Chinese EMRs, achieving an F1-score of 85.78%. However, the reliance on a large medical dictionary and the model's architectural complexity limit its portability and scalability beyond Chinese-language datasets.

Table 1 presents a summary of the reviewed Named Entity Recognition (NER) research works.

**Table 1**. Summary of the reviewed Named Entity Recognition (NER) research works.

| Author | Year | Domain | Pros | Cons |
|---|---|---|---|---|
| Ashrafi et al. [2] | 2020 | General Bangla NER | Early cost-sensitive BiLSTM-CRF; handles imbalance; open-source | Only used BERT based model |
| Ge et al. [3] | 2020 | Medical NER + FL | Privacy preserving; uses cross-platform labeled data; handles heterogeneous annotations | Lower accuracy; high communication cost; early FL design |
| Gligic et al. [7] | 2020 | EHR NER (English) | Outperforms traditional rule-based systems; high dropout rates | Low performance on the least-annotated classes; sparse training samples impacts relation extractions |
| Perera et al. [4] | 2020 | Biomedical NER+RE | Useful joint NER+RE model | Partly survey-like; no longer SOTA (Check) |
| Catelli et al. [6] | 2020 | Italian Clinical De-ID | Effective cross-lingual transfer for low-resource Italian | Italian-only; limited COVID-era data |
| Miranda-Escalada et al. [5] | 2020 | Cancer NER + normalization | First Spanish oncology text mining task;Gold-standard Cantemist corpus | Spanish only; Small dataset; weak prediction result with rare tumors |
| Liu et al. [8] | 2021 | Medical NER (Chinese) | Domain-specific pretraining; large gains over BERT | Chinese-focused; limited generalization |
| French & McInnes [9] | 2023 | Biomedical Entity Linking (survey) | indicates progression from rule-based to current models | English-centric; survey-focused |
| Haque et al. [10] | 2023 | General Bangla NER | Largest Bangla NER dataset; diverse sources | General-domain only; annotation issues; outdated baselines |
| Muntakim et al. [11] | 2023 | Bangla Medical NER | First gold-standard Bangla medical NER; expert-labeled | Very small; few entity types; weak baselines |
| Lima et al. [12] | 2023 | General Bengali NER | Hybrid approach; high (94%) F1 | Dataset heavily skewed; only non-contextualized word embeddings |
| Shahgir et al. [13] | 2023 | Bangla Nested NER | First work on Bangla CoNER, evaluating both CRF and transformer fine-tuning; BanglaBERT achieves the best performance (0.79 F1) | Small dataset; translated corpus |
| Aurpa et al. [14] | 2024 | Bangla Math Entity Dataset | First math entity dataset; well-annotated | Niche; small (20k sentences) |
| Aurpa & Ahmed [15] | 2024 | Bangla MER | Bangla Math Entity dataset, Strong ensemble MER (94–96% F1) | Math-only dataset |
| Farhan et al. [16] | 2024 | BanglaBERT | Proposed a hybrid model combining BanglaBERT embeddings, gazetteer features, and CRF | Gazetteer-dependent |
| García-Barragán et al. [17] | 2025 | Spanish Medical NER | High LLM performance (GPT-4 level) | High cost; latency; hallucination risk; closed-weight |
| Yan et al. [18] | 2025 | Chinese Medical NER | Entity-association + gating; new SOTA | Chinese only; complex architecture |
| Wei et al. [19] | 2025 | General Medical NER | Efficiency-focused; strong reported results | Very recent;limited external validation |
| Sun et al. [20] | 2025 | Medical NER | Knowledge-enhanced + positional encoding; strong gains | Complex; requires external knowledge base |
| Proposed Method | 2025 | Medical NER | Implementation of ensemble transformer models for handling long sequences. | High computational cost, complex architecture, language-specific model |

## 3 Methodology

The following system architecture, as shown in Fig 1, represents a robust framework for Bangla medical entity recognition, leveraging a Multi-BERT ensemble approach. Different parts of this figure are explained further.

The following steps are executed sequentially:

### 3.1 Dataset preparation

We have curated a comprehensive dataset specifically designed for Bangla Medical Entity Recognition (MER), comprising **6,895 observations**. Each observation consists of a medical statement along with annotated medical entities, which are classified into six distinct categories, as summarized in Table 2. For readability, Table 2 presents only the English translations of the original Bangla text; the complete Bangla dataset is available at [21].

The dataset was constructed using real-world medical statements, where medical entities were systematically extracted and labeled. This unique dataset provides a robust foundation for advancing Bangla-language medical text analysis and entity recognition tasks. The pie chart in Fig 2 visually represents the distribution of different medical entity categories based on their count.

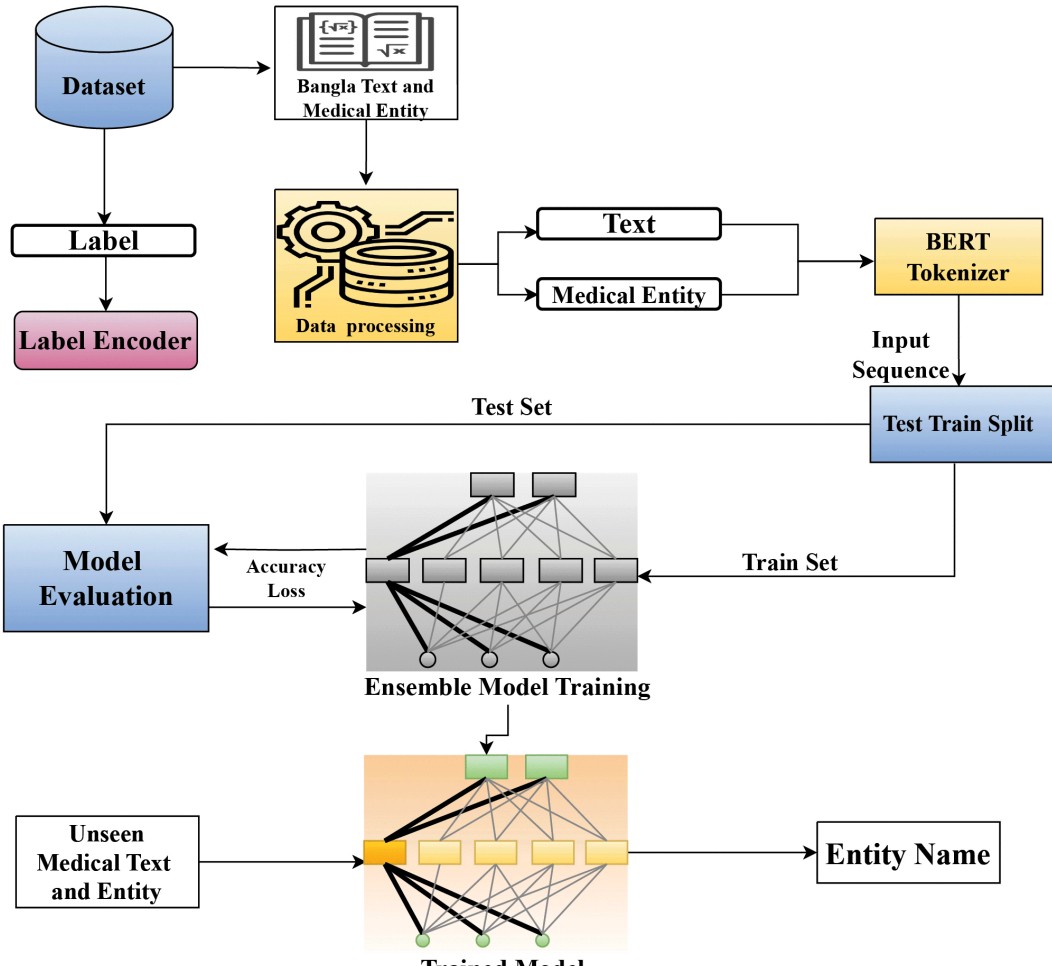

**Fig 1**. System architecture.

**Table 2**. Sample Bangla-MedER Entries (English Translation).

| Language | Text | Entity Name | Entity Type |
|---|---|---|---|
| English Translation | Fexofenadine treats allergy symptoms, including itching, sneezing, and hives. These symptoms include watery eyes, runny nose, and itchy eyes and nose. It works by blocking your body from making natural histamines when you experience an allergic reaction. | Fexofenadine | Medicine/ Chemical Name |
| English Translation | Azithromycin belongs to a group of drugs called macrolide antibiotics, which work by killing bacteria that cause infections. Azithromycin is intended to treat infections caused by bacteria, including but not limited to respiratory tract infections, skin and soft tissue infections, sexually transmitted diseases, and even certain types of pneumonia. | Skin, respiratory system | Organ |
| English Translation | Albendazole, a potent antiparasitic drug, has attracted attention for its effectiveness in treating various worm infestations. This versatile drug can affect both intestinal and tissue parasites. | Worm infestations | Disease |
| English Translation | Aceclofenac belongs to the category of non-steroidal anti-inflammatory drugs (NSAIDs). This medicine is prescribed for chronic inflammation and pain in the bones or joints. Aceclofenac works by blocking the action of an enzyme in the body known as "cyclooxygenase (COX)". This enzyme releases chemical prostaglandins at the site of injury and results in swelling, pain and inflammation. By blocking the COX enzyme, aceclofenac helps to relieve pain and reduce inflammation. | Prostaglandins | Hormones |
| English Translation | Chlorpheniramine is an antihistamine medication used to treat a variety of allergic reactions, including seasonal allergies, allergic rhinitis, hives, and allergic conjunctivitis. It works by blocking the effects of histamine, a chemical that causes allergic reactions in the body. Chlorpheniramine is also effective for treating common cold symptoms such as sinus infections and sneezing. | Antihistamine | Pharmacological Class |
| English Translation | Diclofenac is a non-steroidal anti-inflammatory drug (NSAID). It is intended to relieve pain and reduce inflammation by eliminating the underlying cause or factors that cause pain in the body. It can be given orally, intravenously (into a vein), rectally (through the anus), or injected subcutaneously (under the skin). It works by inhibiting prostaglandin enzymes, which are the cause of pain and inflammation. | Pain reliever | Common Medical Terms |

The largest category is Medicine/Chemical Names, with 1,938 entities, meaning many drug and chemical-related terms exist. Common Medical Terms comes next with 1,127 entities, covering general medical words. Diseases (1,098 entities) and Organ Names (1,066 entities) also have large portions, showing the importance of identifying illnesses and body parts. Pharmacological Classes (877 entities) and Hormones (807 entities) have fewer terms but are still important. The different colors in the chart help compare the categories quickly.

Table 3 contains observations from all classes of medical entities. For each class, we have given the Bangla example and the corresponding English translation.

## 3.2 Data preprocessing

The raw text contains different unusual letters or terms, which may lead to decreased performance. In order to prepare the model for accurate prediction, it is essential to remove this item from the text before training the model. Our Bangla medical text dataset is preprocessed to clean the data and encode entity names for accurate analysis. The following steps outline our preprocessing approach:

• **Eliminating Unnecessary Characters:** In the initial phase of data preprocessing, we focus on cleaning the dataset by removing any extraneous or irrelevant characters, such as symbols ($, %, #, -, etc.), punctuation marks, or other non-textual elements. These characters do not contribute to the semantic meaning of the text and may introduce noise into

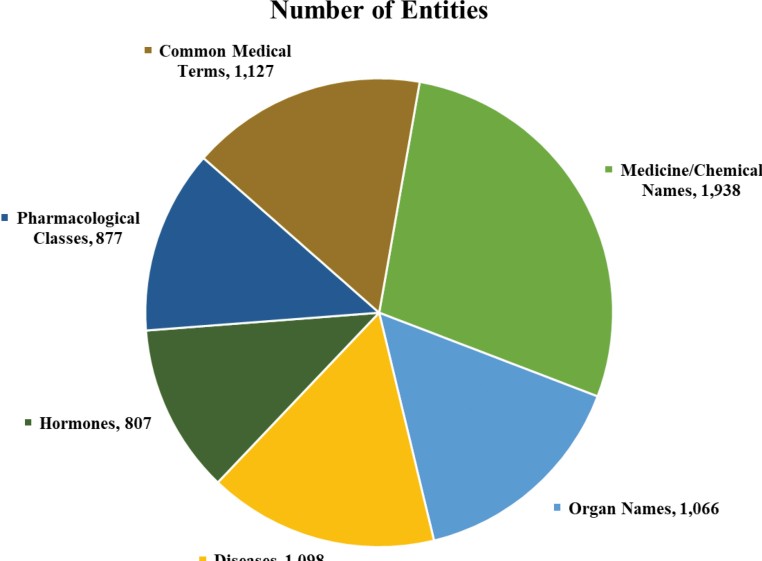

**Number of Entities**

**Fig 2**. **Indicating the number of observations per entity.**

**Table 3**. **Class-wise and overall evaluation metrics for MedER task.**

| Metric | Precision (%) | Recall (%) | F1-Score (%) |
|---|---|---|---|
| **Medicine Chemical Name** | 91.67 | 91.67 | 91.67 |
| **Disease** | 88.55 | 88.55 | 88.55 |
| **Common Medical Terms** | 84.87 | 84.87 | 84.87 |
| **Hormone** | 75.00 | 75.00 | 75.00 |
| **Pharmacological Class** | 87.27 | 87.27 | 87.27 |
| **Organ** | 93.94 | 93.94 | 93.94 |
| **Macro Avg** | 86.55 | 86.55 | 86.55 |
| **Weighted Avg** | 88.15 | 88.15 | 88.15 |
| **Overall Accuracy** | 89.58 | | |
| **Micro F1-Score** | 89.58 | | |
| **Macro F1-Score** | 87.67 | | |

the dataset, potentially hindering the model's ability to learn meaningful patterns. By eliminating such characters, we ensure that the input data is clean, structured, and more suitable for analysis.

- **Removing Common Bangla Stop Words:** To further refine the dataset and enhance the model's accuracy, we filter out common Bangla stop words. These words often appear frequently in text, but carry minimal semantic weight in the context of natural language processing (NLP) tasks such as sentiment analysis or topic modeling. Removing stop words reduces the dimensionality of the dataset and allows the model to focus on more relevant and meaningful terms that contribute to understanding the content.

- **Applying Stemming and Lemmatization for Normalization:** Bangla words can appear in multiple inflected or derived forms depending on their grammatical use, which can lead to redundancy in text processing. We apply stemming and lemmatization techniques to normalize these variations by reducing words to their root or base forms. **Stemming** focuses on removing affixes (e.g., suffixes or prefixes), while **lemmatization** uses linguistic rules to ensure the base form is meaningful and grammatically correct. These techniques improve data consistency, reduce redundancy, and enhance the model's ability to generalize across diverse forms of the same word.

The entire data preprocessing techniques are shown in Fig 3.

### 3.3 Tokenization

After cleaning and normalizing the text, the dataset is tokenized using the **BERT tokenizer** [22,23], which splits the text into subword units and generates input sequences tailored for classification tasks. This step is critical for leveraging pre-trained BERT models, as it ensures compatibility with the model's expected input format and retains the semantic integrity of the text. As we were assembling two different BERT layers, we created different input sequences for each BERT. For BERT layer one, the [SEP] token follows the text, and entities are added after the [SEP] token. We have created the opposite combination for the input sequence for the BERT layer two. Then, we tokenize these sequences to pass them to the model.

### 3.4 Model training

The input sequences, generated during the preprocessing and tokenization stages, are utilized as features for training both the BERT and the ensemble BERT models. Each sequence is paired with its corresponding encoded labels, which denote the medical entity categories, such as medicine, disease, organ, and others. These labeled pairs enable the models to learn and classify the entities accurately. Model performance is assessed using test data, with metrics like accuracy and loss, while hyperparameter tuning is applied for optimization.

   **3.4.1 Experimental setup.** To obtain GPU and TPU resources, we worked with Google Colab. An NVIDIA Tesla K-80 GPU with 12 GB of RAM was used for the research. Colab facilitates the seamless execution of transformer model by providing a Python runtime with pre-installed libraries and packages.

   **3.4.2 Hyperparameter tuning.** To identify the optimal hyperparameter configuration for our models, we experimented with various values for learning rate, batch size, maximum sequence length, and number of epochs. The hyperparameters provide the best outcome for the proposed Multi-BERT Ensemble model are learning rate = 2e-4, batch size = 32, maximum sequence length = 484, and 40 epochs. The AdamW optimizer was used, and setting verbose = 1 enabled convenient tracking of the training process.

   **3.4.3 Input sequence generation.** To capture linguistic and positional information in the text, the following feature representations are used:

1. Word Embedding: Pretrained word embeddings for Bangla are utilized to map each token into a high-dimensional vector representation, capturing semantic relationships.
2. Segment Encoding: Encodes the sequence information to differentiate between different parts of the text, enabling the model to distinguish between tokens belonging to different segments.
3. Positional Encoding: Positional encodings are added to the embeddings to introduce the notion of word order, which is essential for processing sequential data.

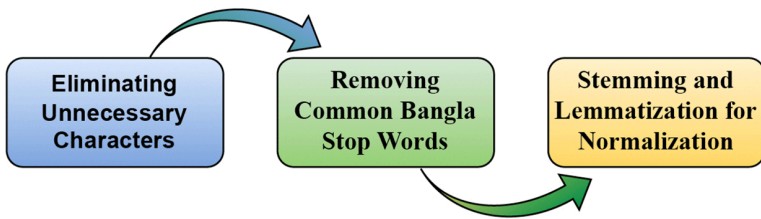

**Fig 3**. Data preprocessing process of MedER.

In this research, we propose an ensemble method for medical entity recognition showed in Fig 4 that leverages input information in diverse ways to improve performance. The BERT model, when processing two text sequences, requires special tokens such as [CLS], [PAD], and [SEP]. The [SEP] token is used to separate the two sequences. For this task, the sequences consist of medical text and the corresponding medical entities. We constructed two distinct combinations of input sequences as follows:

- First Sequence: Tokens of the medical text are placed after the [CLS] token, followed by the [SEP] token, and then the tokens of the medical entity. The sequence is padded with the [PAD] token as needed.
- Second Sequence: Tokens of the medical entity are placed before the [SEP] token, followed by the tokens of the medical text after it.

These two input sequences are tokenized and passed to separate BERT layers, each processing one sequence independently. The outputs from the two BERT layers are then combined using a concatenation layer to capture diverse contextual representations. The concatenated output is processed through a fully connected layer, and the final classification is performed by a softmax layer, which predicts the probability distribution over medical entity classes. Fig 4 illustrates the

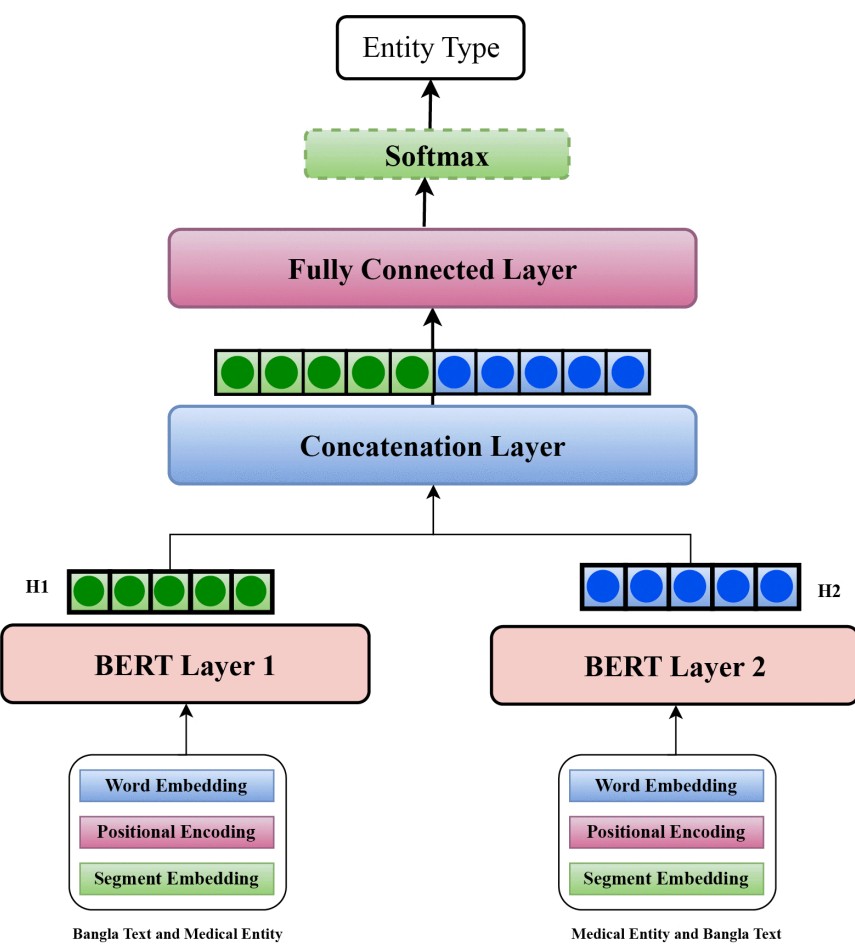

**Fig 4**. Proposed-model.

architecture of the proposed ensemble model, highlighting how raw text is tokenized, attention masks are generated, and predictions are made through this multi-layered approach.

### 3.4.4 Model architecture.
The model architecture consists of the following key components:

- **E**nsemble BERT Layers: The enriched embeddings are passed through two separate BERT layers (BERT Layer 1 and BERT Layer 2) to extract contextual representations of the text. Each BERT layer captures deep semantic and syntactic features by attending to all tokens in the input sequence. An ensemble strategy is employed to combine the outputs of the two BERT layers for improved representation.
- **C**oncatenation Layer: The outputs from the two BERT layers are concatenated to form a unified representation. This step ensures that information from both layers is effectively aggregated.
- **F**ully Connected Layer: The concatenated representation is fed into a fully connected layer to map the features to a lower-dimensional space, preparing them for classification. Nonlinear activation functions are applied to introduce non-linearity and improve the model's expressive power.
- **S**oftmax Layer: The final output layer employs the softmax activation function to predict the probability distribution over the possible entity categories. This enables the identification of specific medical entities in the input Bangla text. An ensemble strategy is employed to combine the outputs of the two BERT layers for improved representation.

## 3.5 Model evaluation

For the evaluation of our model and comparison of its performance with other approaches, we have discovered various evaluation matrices. At first, we determined the TP (True Positive) and TN (True Negative), which are the right predictions, and FP (False Positive) and FN (False Negative), the wrong predictions of the model from the confusion matrix. Using these measures, we found accuracy to be one of the most effective and popular evaluation scores with the help of Eq 1.

$$\text{Accuracy} = \frac{TP + TN}{TP + TN + FP + FN} \tag{1}$$

Next, using the Eqs 2 and 3, the Precision and Recall are determined.

$$\text{Precision} = \frac{TP}{TP + FP} \times 100\% \tag{2}$$

$$\text{Sensitivity/Recall} = \frac{TP}{TP + FN} \times 100\% \tag{3}$$

These two values are utilized in Eq 4 for determining the F1 Score, and the F1 Score is used for finding the Macro average F1 score in Eq 5.

$$\text{F1 Score} = 2 \star \frac{\text{Precision} \star \text{Recall}}{\text{Precision} + \text{Recall}} \tag{4}$$

$$\text{Macro FI Score} = \frac{\sum_{i=1}^{\text{No of classes}} \text{F1 Score}_i}{\text{No of classes}} \tag{5}$$

Moreover, we have found out the Micro F1 score with the help of Eq 6.

$$\text{Micro F1 Score} = \frac{\sum TP}{\sum TP + \frac{1}{2}\left(\sum FN + \sum FP\right)} \tag{6}$$

Finally, we measured the weighted average and macro average of all the evaluation metrics using Eqs 7 and 8.

$$\text{Weighted Average} = \sum_{i=1}^{N} \frac{n_i}{n} \times \text{Metric}_i \tag{7}$$

$$\text{Macro Average} = \frac{1}{N} \sum_{i=1}^{N} \text{Metric}_i \tag{8}$$

**Where:**

- $N$ is the number of classes.
- $n_i$ is the total number of instances in $i$ class.
- $n$ is the number of total instances.
- $\text{Metric}_i$ is the precision, recall, or F1-score for class $i$.

## 4 Result

### 4.1 Performance of the proposed model

The preprocessed Bangla text and entity classes are used to train the proposed ensemble architecture. To analyze the performance, we have determined the confusion matrix of our model and represented that as a heatmap in Fig 5. The confusion matrix assists evaluate the performance of the proposed MedER model. The matrix compares the actual labels (on the left) with the predicted labels (on the bottom). Each cell in the matrix tells us how many times a specific class was predicted correctly or incorrectly. The color scale highlights the frequency of predictions, with red showing high numbers and dark blue indicating fewer cases. The diagonal values represents correct predictions. The model performed best on "Medicine/Chemical Name" correctly classifying 220 cases, followed by Disease with 116 correct classifications. It also identified 101 cases correctly for "Common Medical Terms," 96 for "Pharmacological Class," and 91 for "Organ". However, it struggled with "Hormone," correctly classifying only 21 cases. The lower accuracy for "Hormone" suggests that the model may have difficulty distinguishing it from other medical-related terms. The off-diagonal values in the confusion matrix represent misclassifications, where the model predicted the wrong category. The most significant misclassification occurred in the "Disease" category, where 14 cases were incorrectly classified as "Common Medical Terms." Similarly, 8 "Common Medical Terms" cases were misclassified as "Disease," showing some overlap between these two categories. The "Medicine/Chemical Name" category had some errors as well, with 8 cases misclassified as "Hormone" and 3 as "Pharmacological Class." This suggests that some medicine-related terms are being confused with Hormone name. The model also misclassified 6 "Pharmacological Class" cases as "Medicine/Chemical Name," likely due to similarities between these categories. In the "Hormone" category, 2 cases were incorrectly classified as "Medicine/Chemical Name" and 2 as "Common Medical Terms," indicating possible overlap in terminology. For "Organ" names, there were fewer misclassifications, with only 3 cases predicted as "Disease" and 2 as "Common Medical Terms." This suggests that "Organ" names are more distinct from other medical entities. Overall, the confusion matrix reveals that the model has strong classification performance but struggles with differentiating between similar medical terms, particularly between "Disease," "Common Medical Terms," and "Medicine/Chemical Name."

In the advocate of our proposed model's remarkable performance, we have measured different evaluation metrics in Table 3. The table presents the evaluation metrics for different categories in the MedER task, showing how well the proposed MedER model performs in identifying different medical entities. The key metrics used are Precision, Recall, and F1-Score, all expressed as percentages.

**Fig 5**. **Confusion matrix of the proposed model.**

- Medicine Chemical Name has the highest performance, with 91.67% in all three metrics, meaning the model correctly identifies and retrieves most of these terms.
- Disease performs well, with an 88.55% F1-Score, showing that the model can recognize diseases with high accuracy.
- Common Medical Terms have a slightly lower F1-Score of 84.87%, indicating that some of these terms might be harder for the model to distinguish.
- The hormone has the lowest scores (75.00% for all three metrics), meaning the model struggles more in identifying hormones correctly.
- The Pharmacological Class has an 87.27% F1-Score, suggesting strong performance in recognizing this category.
- Organ achieves the highest accuracy among individual categories, with an F1-Score of 93.94%, showing that organ names are well-identified.

The model's overall performance is measured using different methods. The Macro Average (86.55%) gives equal importance to all categories and shows the general performance. The Weighted Average (88.15%) considers the number of examples in each category, making it a more balanced measure. The Overall Accuracy (87.87%) tells us how many predictions were correct out of all cases. The Micro F1-Score (87.87%) looks at total correct predictions across all categories, which helps when some categories have more examples than others. The Macro F1-Score (86.55%) is similar to the

Macro Average but focuses on both precision and recall. Together, these numbers help us understand how well the model is performing.

## 4.2 Comparative analysis of the proposed model

We compared different transformer models to see how well they performed on a dataset. The bar chart in Fig 6 shows their accuracy, with BERT-based models doing better than others. Our model, Ensemble BERT, had the highest accuracy at 89.58%, proving that it is the best at handling this dataset. This result shows that our approach is strong and reliable. The next best model was BERT, with 77.78% accuracy. While it performed well, it was still much lower than Ensemble BERT. DistillBERT had an accuracy of 57.89%, showing a clear drop in performance. This model is a smaller and faster version of BERT, but it sacrifices some accuracy. ELECTRA scored 55.54%, even though it uses a different training method. It did not perform as well as the top models. RoBERTa had the lowest accuracy at 51.67%. While this model works well in other cases, it was not very effective on this dataset. Except for the BERT and Ensemble BERT model, other transformers have provided inferior results on the dataset because they are not trained in Bangla. Overall, our Ensemble BERT model was the best by a large margin. This proves that our method is more advanced and effective compared to other models.

As BERT is pretrained in Bangla, we will enhance the analysis of the changes in performance with and without ensembling. Fig 7 shows us the improved performance of the ensemble multi-layer BERT model over a single BERT model. The bar graph in Fig 7 shows that Ensemble BERT performs significantly better than BERT across all evaluation metrics. Let's break down the improvements in percentage points for each metric: Accuracy increased from 77.78% to 89.58%, an improvement of 11.8 percentage points. Micro Average F1 Score also improved from 77.78% to 89.58%, a gain of 11.8 percentage points. Macro Average F1 Score rose from 74.66% to 87.67%, showing an increase of 13.01 percentage points. Precision went up from 80.3% to 91.49%, improving by 11.19 percentage points. Recall saw the highest

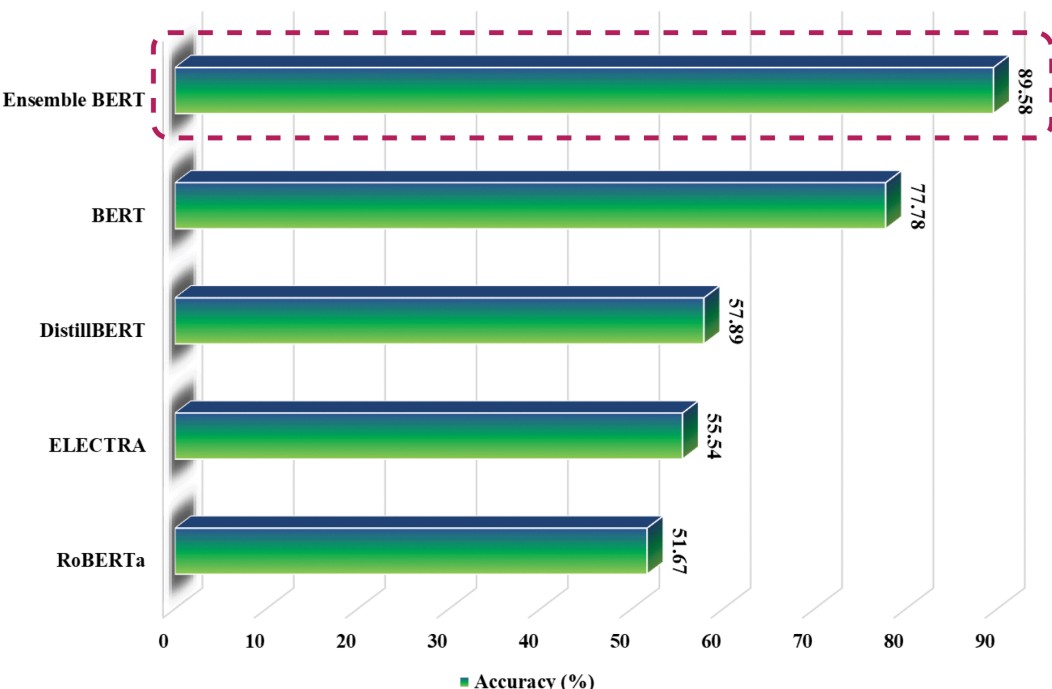

**Fig 6**. Accuracy of different transformer models on the our data.

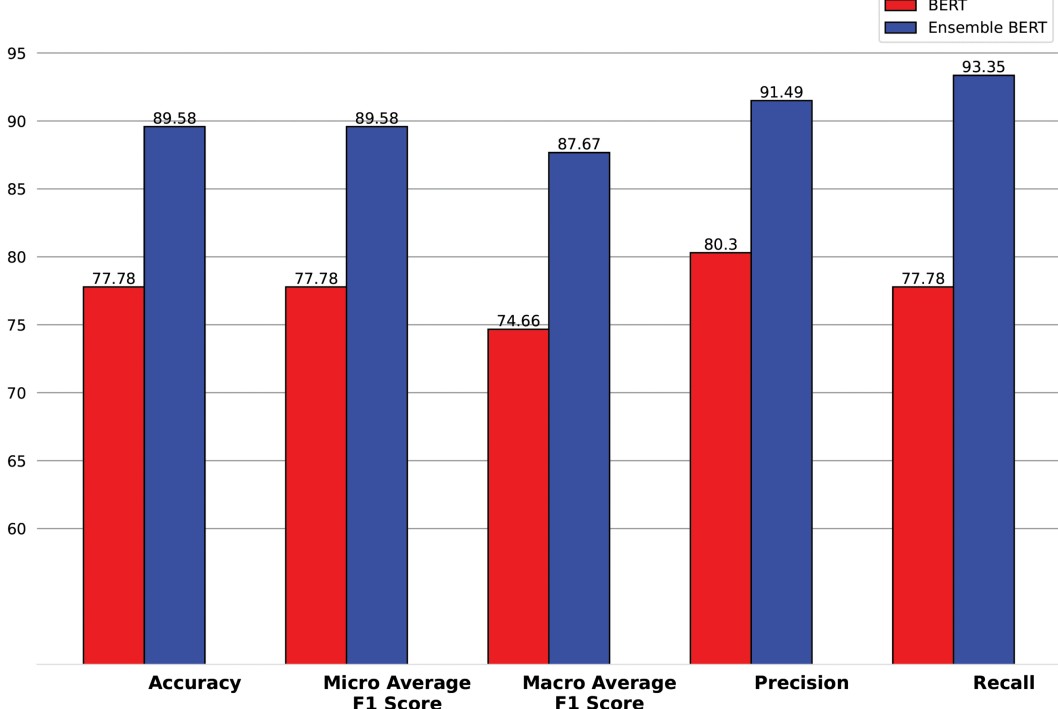

**Fig 7**. Different evaluation metrics of One layer BERT vs Multi layer BERT.

jump, from 77.78% to 93.35%, increasing by 15.57 percentage points. These improvements show that Ensemble BERT is much more accurate, precise, and reliable than the basic BERT model. The most significant improvement is in Recall (+15.57%), meaning Ensemble BERT is much better at correctly identifying all relevant cases.

## 5 Discussion

Our study focused on the Bangla MedER task, where we tested different transformer-based models to recognize medical terms. The results gave us valuable insights into how well our models performed, how our dataset was structured, and what we could improve. The Bangla MedER dataset can play a significant role in Bangla NLP research. The dataset is well-balanced, covering different medical categories such as Medicine/Chemical Names, Organ Names, Diseases, Common Medical Terms, Pharmacological Classes, and Hormones. The largest category, Medicine/Chemical Names, has 1,938 entities, while Organs and Diseases have over 1,000 entities each. This balance helps the model learn effectively and improves its accuracy.

The confusion matrix and evaluation scores show that the model does a great job at identifying Medicine/Chemical Names, Diseases, and Organs with high accuracy. However, the Hormones category had slightly lower scores, meaning the model struggles more with this type of term. Overall, the high precision, recall, and F1 scores show that our model is strong and performs well.

When we compared different transformer models, Ensemble BERT stood out as the best, achieving 89.48% accuracy, which is much higher than the other models. BERT was the second-best but still had a big performance gap compared to Ensemble BERT. Meanwhile, DistilBERT, ELECTRA, and RoBERTa had lower accuracy, meaning they were not as effective for this task. Except for the BERT, other models are not trained in Bangla. Unlike BERT, other transformer models can not capture the complex text patterns of the Bangla language. We also tested how model depth affects performance

to advocate why we have increased the BERT layers. Multi-layer BERT (Ensemble BERT) performed much better than One-Layer BERT in all key metrics, such as accuracy, precision, recall, and F1 scores. The accuracy score improved by 11. This shows that using multiple layers helps the model understand more complex patterns, leading to better results. It actually preserves input features, Text, and Entity names for long sequences.

Our work on Bangla MedER has a significant real-world impact, particularly in the medical field. Accurately identifying medical terms from Bangla text helps healthcare professionals process medical documents, improve electronic health records, and support research by quickly extracting important terms. This system can help create AI-powered medical chatbots, making healthcare data more useful for building automated Bangla medical systems. It also improves machine translation and other language processing applications, ensuring medical texts are understood more accurately. Bangla MedER helps close the language gap in Bangla medical AI, giving useful tools to Bangla-speaking communities. This research can improve healthcare, medical research, and technology by making medical information more transparent, accessible, and valuable.

## 6 Conclusion

This research focuses on leveraging BERT for Medical Entity Recognition (MEdR) in Bangla medical texts, addressing key challenges such as limited annotated datasets and complex language variations. MEdR, an advanced form of Named Entity Recognition (NER), plays a crucial role in the healthcare industry by identifying medical entities like medicine names, diseases, and medical terms in unstructured texts. Using a newly developed dataset of 6,895 Bangla medical statements, we fine-tune BERT to accurately classify these entities. This work not only fills the gap in Bangla medical NLP resources but also has practical applications in automating record management, supporting diagnoses, and improving healthcare accessibility for Bangla-speaking communities, ultimately enhancing research productivity and decision-making in healthcare. Experimental results demonstrate that our proposed Multi-BERT Ensemble model achieves 89.58% accuracy and a macro F1-score of 87.87%.

### 6.1 Contributions

The results of this study offer a number of useful insights. Our approach can use in identifying and categorizing key information from unstructured medical text in a structured way. This can help our physicians in making decisions and improving patient cares. Additionally, this structured data can be used in developing medical assistance particularly for Bangla-speaking people. It also helps in medical research.

### 6.2 Challenges

The proposed MultiBERT approach enhances model performance, but it's computationally heavy. The model requires a large amount of processing power and makes it more challenging to carry out rapid experiments. Bangla medical texts can sometimes be challenging because of code-mixed English terminology, uneven spellings, abbreviations, and colloquial idioms. The lack of domain-specific pretrained Bangla biomedical models also affects semantic comprehension.

### 6.3 Limitations

There are a few constraints to this work. Despite having significant medical content, the dataset's size is limited. Because it can only recognize entities relevant to medicine, its real-time use is confined.

### 6.4 Future work

We plan to improve our Bangla MedER project in the future. First, we intend to expand our dataset by adding more diverse medical terms to make the model more accurate. We will also fine-tune our existing transformer models and

explore more advanced transformer architectures to boost performance. To improve our results further, we will use cross-lingual transfer learning, which means learning from other languages to enhance our Bangla model. Conducting a detailed error analysis will help us understand where our model struggles, and developing ensemble methods (combining multiple models) will help improve accuracy. Another important goal is to make our model more interpretable so users can understand and trust its predictions. We also plan to work closely with medical experts to refine our dataset and ensure that our model is practically helpful in real-world applications. By making these improvements, we hope to advance natural language processing in the medical field for the Bangla-speaking community, making medical information more accessible and accurate.

## Code availability

The source code is available at https://github.com/Taharat22/Medical-Entity-Recognition. The annotated Bangla MedER dataset is publicly available on Kaggle Data Source: https://www.kaggle.com/datasets/tanjimtaharataurpa/bangla-medical-entity-dataset.

## Acknowledgments

The author sincerely appreciates NXTLab - Next-Generation Technologies Lab, UFTB for providing essential technical support and access to infrastructure, which facilitated the successful completion of this study

## Author contributions

**Conceptualization:** Tanjim Taharat Aurpa, Farzana Akter.

**Data curation:** Md. Mehedi Hasan, Shakil Ahmed, Rubel Sheikh.

**Formal analysis:** Rubel Sheikh.

**Methodology:** Tanjim Taharat Aurpa, Farzana Akter, Rubel Sheikh.

**Validation:** Md. Mehedi Hasan, Shakil Ahmed, Rubel Sheikh.

**Visualization:** Md. Mehedi Hasan, Rubel Sheikh.

**Writing – original draft:** Tanjim Taharat Aurpa, Rubel Sheikh.

**Writing – review & editing:** Farzana Akter, Shifat Ara Rafiq, Fatema Khan, Rubel Sheikh.

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
