## [Decision Letter · Decision Letter 0]

31 Oct 2025

PONE-D-25-12318Bangla MedER: Multi-BERT Ensemble Approach for the Recognition of Bangla Medical EntityPLOS ONE

Dear Dr. Aurpa,

Thank you for submitting your manuscript to PLOS ONE. After careful consideration, we feel that it has merit but does not fully meet PLOS ONE’s publication criteria as it currently stands. Therefore, we invite you to submit a revised version of the manuscript that addresses the points raised during the review process.

As you can see the reviewers and the editor are suggesting a revise due to aspects related to related work, comparison and PLOS ONE's compliance on data and code.

We look forward to receiving your revised manuscript.

Kind regards,

Issa Atoum

Academic Editor

PLOS ONE

Journal Requirements:

““The author sincerely appreciates NXTLab - Next-Generation Technologies Lab, BDU for providing essential facilities and resources that significantly contributed to thesuccessful completion of this research””

4. Please note that your Data Availability Statement is currently missing [the repository name and/or the DOI/accession number of each dataset OR a direct link to access each database]. If your manuscript is accepted for publication, you will be asked to provide these details on a very short timeline. We therefore suggest that you provide this information now, though we will not hold up the peer review process if you are unable.

Additional Editor Comments:

Kindly include these in your revised manuscript.

Include quantitative impact in both the abstract and conclusion, and add a subsection addressing limitations, challenges, and contributions. Summarize the literature review in a comparative table, provide more details on datasets and experimental setup, and increase the number of references to strengthen the scholarly foundation. Clearly indicate how data were extracted, ensure compliance with PLOS ONE’s code and dataset availability requirements. Despite the inclusion of Section 7 on ethical statements, further enhancements are required for PLOS ONE compliance include clear statement on ethical considerations related to data collection and usage.

Reviewers' comments:

Reviewer's Responses to Questions

**Comments to the Author**

1. Is the manuscript technically sound, and do the data support the conclusions?

Reviewer #1: Yes

Reviewer #2: Yes

2. Has the statistical analysis been performed appropriately and rigorously?

Reviewer #1: Yes

Reviewer #2: Yes

3. Have the authors made all data underlying the findings in their manuscript fully available?

Reviewer #1: Yes

Reviewer #2: Yes

4. Is the manuscript presented in an intelligible fashion and written in standard English?

Reviewer #1: Yes

Reviewer #2: Yes

5. Review Comments to the Author

Reviewer #1: This paper proposes a novel MultiBERT Ensemble approach for MedER in Bangla text. this method utilizes an ensemble of multi-layer BERT models to recognize medical entities in Bangla medical text, offering a unique contribution to the field. The proposed model outperforms existing transformer-based models, achieving an 11.80% accuracy improvement over a singlelayer BERT model. for low-resource languages and lacking of annotated datasets a high-quality dataset tailored for the Bangla MedER task is developed. The dataset was used to evaluate the effectiveness of the model through multiple performance metrics, demonstrating its robustness and applicability. The findings highlight the potential of Multi-BERT Ensemble models in improving MedER for Bangla and set the foundation for further advancements in low-resource medical NLP..

Good work keep up

But some comments is needed and submitted to he editor

Reviewer #2: 1- The introduction is very simple and needs further expansion to clarify the research gap and what researchers in this field are seeking.

2- Sort research papers in Related Work by year of publication (from oldest to newest) and add research papers published in 2025. It is also preferable to add a table summarizing the pros and cons of each included research paper.

3- Comparison of the submitted work with other research papers published in the years 2023-2025

4- Adding recent references published in 2025, with no less than three references

6. PLOS authors have the option to publish the peer review history of their article (what does this mean?). If published, this will include your full peer review and any attached files.

Reviewer #1: No

Reviewer #2: No

---

## [Author Response · Author response to Decision Letter 1]

31 Dec 2025

Response to the Reviewer:

Editor

1. Include quantitative impact in both the abstract and conclusion, and add a subsection addressing limitations, challenges, and contributions.

2. Response: We have addressed this comment as follows:

• The quantitative impact has been highlighted in both the abstract and conclusion. For example, we explicitly state: “The proposed model outperforms existing transformer-based models, achieving an 11.80% accuracy improvement over a single-layer BERT model.”

• New subsections titled 6.1 Contributions, 6.2 Challenges and 6.3 Limitations have been added to clearly discuss the limitations of our approach, the challenges faced during development, and the key contributions of our work.

3. Summarize the literature review in a comparative table, provide more details on datasets and experimental setup, and increase the number of references to strengthen the scholarly foundation.

Response: The Related Work section has been updated accordingly. Details about datasets and experimental setup in 3.4 Model Training subsection have been updated.

4. Clearly indicate how data were extracted, ensure compliance with PLOS ONE’s code and dataset availability requirements.

Response: Details about dataset availability have been mentioned in section 7 Data Availability.

5. Despite the inclusion of Section 7 on ethical statements, further enhancements are required for PLOS ONE compliance include clear statement on ethical considerations related to data collection and usage.

Response: Details about dataset availability and ethical considerations related to data collection and usage have been mentioned in section 7 Data Availability.

Reviewer 2

1. The introduction is very simple and needs further expansion to clarify the research gap and what researchers in this field are seeking.

Response: The Introduction section has been updated accordingly.

2. Sort research papers in Related Work by year of publication (from oldest to newest) and add research papers published in 2025. It is also preferable to add a table summarizing the pros and cons of each included research paper.

Response: The Related Work section has been updated accordingly. Papers are now sorted chronologically from old to newest, and a table summarizing the pros and cons of each included research paper has been added.

3. Comparison of the submitted work with other research papers published in the years 2023-2025

Response: The manuscript has been revised to include a comparison of the submitted work with relevant research published between 2023 and 2025.

4. Adding recent references published in 2025, with no less than three references

Response: Four references from 2025 have been incorporated into the manuscript in Literature Review Section.

---

## [Decision Letter · Decision Letter 1]

12 Jan 2026

PONE-D-25-12318R1Bangla MedER: Multi-BERT Ensemble Approach for the Recognition of Bangla Medical EntityPLOS One

Dear Dr. Akter,

Thank you for submitting your manuscript to PLOS ONE. After careful consideration, we feel that it has merit but does not fully meet PLOS ONE’s publication criteria as it currently stands. Therefore, we invite you to submit a revised version of the manuscript that addresses the points raised during the review process.

**Upon internal review, we require the following revisions to align the manuscript with PLOS ONE’s Data Availability Policy. Please address these points to ensure the transparency and reproducibility of your findings:**

**Code Sharing & Reproducibility:** In accordance with our reproducibility guidelines, please provide the specific source code or scripts used to generate the results presented in Table 5 and Figures 6 and 7. We recommend depositing this code in a public repository (e.g., GitHub, Zenodo, or Figshare) and providing the DOI or link within your Data Availability Statement.

**Data Alignment:** There appear to be discrepancies between the figures and their corresponding tables. Please ensure that all data points are fully synchronized across Table 5, Figure 6, and Figure 7.

**Comprehensive Metrics:** The current manuscript does not provide a complete set of class label metrics for the top-performing model. Please update the results section to include a granular breakdown (e.g., precision, recall, and F1-score for each class) to allow for a thorough validation of the model's performance.

We look forward to receiving your revised manuscript.

Kind regards,

Issa Atoum

Academic Editor

PLOS One

**Journal Requirements:**

Reviewers' comments:

Reviewer's Responses to Questions

**Comments to the Author**

1. If the authors have adequately addressed your comments raised in a previous round of review and you feel that this manuscript is now acceptable for publication, you may indicate that here to bypass the “Comments to the Author” section, enter your conflict of interest statement in the “Confidential to Editor” section, and submit your "Accept" recommendation.

Reviewer #2: (No Response)

2. Is the manuscript technically sound, and do the data support the conclusions?

Reviewer #2: (No Response)

3. Has the statistical analysis been performed appropriately and rigorously?

Reviewer #2: (No Response)

4. Have the authors made all data underlying the findings in their manuscript fully available?

Reviewer #2: (No Response)

5. Is the manuscript presented in an intelligible fashion and written in standard English?

Reviewer #2: (No Response)

6. Review Comments to the Author

Reviewer #2: (No Response)

7. PLOS authors have the option to publish the peer review history of their article (what does this mean?). If published, this will include your full peer review and any attached files.

Reviewer #2: No

---

## [Author Response · Author response to Decision Letter 2]

24 Jan 2026

ID: PONE-D-25-12318R1

Title: Bangla MedER: Multi-BERT Ensemble Approach for the Recognition of Bangla Medical Entity

Code Sharing & Reproducibility: In accordance with our reproducibility guidelines, please provide the specific source code or scripts used to generate the results presented in Table 5 and Figures 6 and 7. We recommend depositing this code in a public repository (e.g., GitHub, Zenodo, or Figshare) and providing the DOI or link within your Data Availability Statement.

Response: The complete implementation can be accessed through the Data and Code Availability section of the manuscript, which provides the corresponding repository link.

Data Alignment: There appear to be discrepancies between the figures and their corresponding tables. Please ensure that all data points are fully synchronized across Table 5, Figure 6, and Figure 7.

Response: There is no Table 5 in the manuscript; the comment likely refers to Table 3. Upon careful review, we identified a minor typographical error which has now been corrected, and the data across Table 3, Figure 6, and Figure 7 are fully synchronized.

Comprehensive Metrics: The current manuscript does not provide a complete set of class label metrics for the top-performing model. Please update the results section to include a granular breakdown (e.g., precision, recall, and F1-score for each class) to allow for a thorough validation of the model's performance.

Response: The class-wise performance metrics for the top-performing model are reported in Table 3. Figure 7 presents an overall performance comparison of the models, offering a complementary summary of the results.

---

## [Editor Report · Decision Letter 2]

27 Jan 2026

Bangla MedER: Multi-BERT Ensemble Approach for the Recognition of Bangla Medical Entity

PONE-D-25-12318R2

Dear Dr. Akter,

We’re pleased to inform you that your manuscript has been judged scientifically suitable for publication and will be formally accepted for publication once it meets all outstanding technical requirements.

Kind regards,

Issa Atoum

Academic Editor

PLOS One
---

## [Editor Report · Acceptance letter]

PONE-D-25-12318R2

PLOS One

Dear Dr. Akter,

I'm pleased to inform you that your manuscript has been deemed suitable for publication in PLOS One. Congratulations! Your manuscript is now being handed over to our production team.

Kind regards,

on behalf of

Dr. Issa Atoum

Academic Editor

PLOS One